# Gait Biomechanics for Fall Prevention among Older Adults

Hanatsu Nagano

Institute for Health and Sport (IHES), Victoria University, Melbourne, VIC 3011, Australia;
hanatsu.nagano@vu.edu.au

**Abstract:** In our currently ageing society, fall prevention is important for better healthy life expectancy and sustainable healthcare systems. While active outdoor walking is recommended as adequate exercise for the senior population, falls due to tripping and slipping exist as the primary causes of severe injuries. Minimum foot clearance (MFC) is the lowest vertical height of the foot during the mid-swing phase and indicates the risk of tripping. In contrast, coefficient of friction (COF) factors determine the occurrence of falls from slipping. Optimisation of the MFC and the COF for every step cycle prevents tripping and slipping, respectively. Even after the initiation of hazardous balance loss (i.e., tripping and slipping), falls can still be prevented as long as the requirements for balance are restored. Biomechanically, dynamic balance is defined by the bodily centre of mass and by the base of support: *spatially*—margin of stability and *temporally*—available response time. Fall prevention strategies should, therefore, target controlling the MFC, the COF and dynamic balance. Practical intervention strategies include footwear modification (i.e., shoe-insole geometry and slip-resistant outsoles), exercise (i.e., ankle dorsiflexors and core stabilisers) and technological rehabilitation (i.e., electrical stimulators and active exoskeletons). Biomechanical concepts can be practically applied to various everyday settings for fall prevention among the older population.

**Keywords:** fall prevention; tripping; slipping; dynamic balance; minimum foot clearance; required coefficient of friction; margin of stability; available response time; gait analysis; biomechanics

## 1. Introduction

Advancements in medical science and social security systems support longevity, and in our modern society, the life expectancies of developed nations are generally above 80 yrs, which has sharply increased since the middle of the last century [1]. While humans have pursued a longer lifespan, if their well-being and quality of life (QOL) are insufficient, not only senior individuals themselves but also the surrounding family could suffer due to the need for constant care and high medical costs. For a country with a severe trend of ageing, such as Japan, healthcare costs (i.e., medical, long-term nursing care, etc.) have already exceeded the total national revenue [2]. As taxation alone is not sufficient, national bonds have been issued as one of the few options to match these increasing financial demands [3,4]. Without accommodating for the fundamental adaptations necessary for an ongoing ageing society, current unsustainable healthcare systems will soon face critical degradation. Instead of mere longevity, our attention should be shifted to the concept of 'healthy life expectancy', broadly defined as "the time in which people can live with sufficient well-being and QOL" [5]. To improve our 'healthy life expectancy', it is important to prevent or even reverse the onset of 'frailty' among older adults.

'Frailty' in the context of ageing refers to general health declines, which can be divided into physical (e.g., sarcopenia), mental (e.g., depression) and social components (e.g., isolation) [6,7]. As illustrated in Figure 1, falls are the devastating result of frailty and a trigger for other health issues [8]. The current review, therefore, focuses on 'falls among senior adults' due to the high frequency and serious consequences of falls (i.e., hospitalisation and injury-related death).

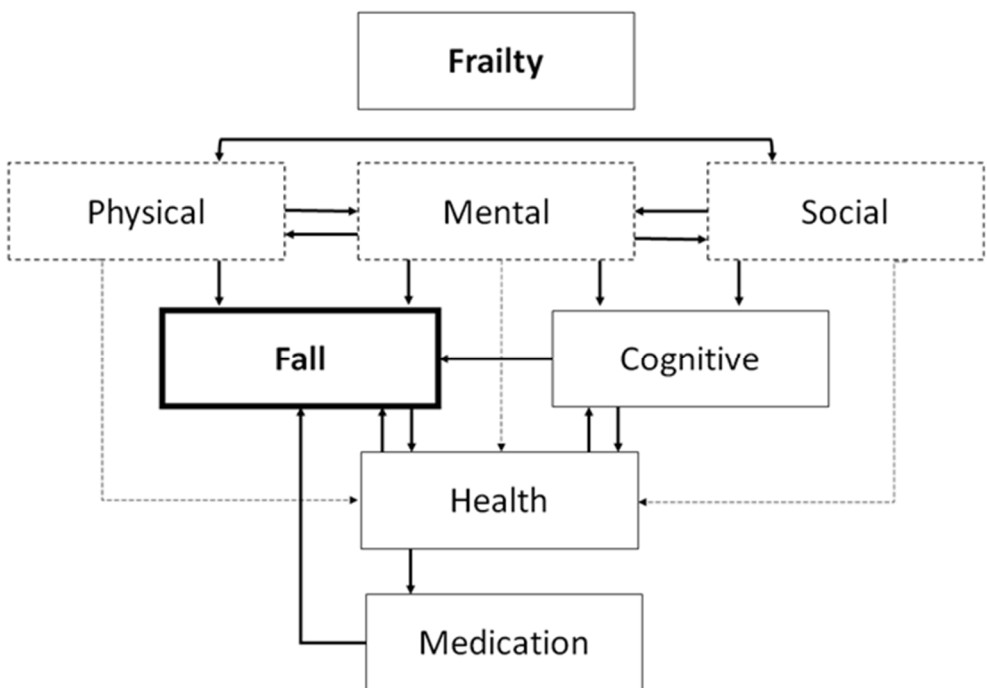

**Figure 1.** Diagram of frailty, falls and associated elements.

Adequate exercise such as 'active walking' is vital and strongly recommended for older adults to maintain their physical, mental and social health, but falls risks paradoxically exist when they are walking [9]. Therefore, it is important to ensure safe walking, but until recently, limited techniques were available to quantify the risk of movement-related falls. As such, only general associations with actual falls episodes have been reported with respect to chronic health conditions, the history of falls, medication (polypharmacy), muscle atrophy (e.g., sarcopenia), mental health, cognitive functions, osteoarthritis, motor control systems, visual and auditory functions, joint range of motion, connective tissue conditions (e.g., tendons and ligaments), proprioception and bone health [10–23]. These findings have been useful for general guidelines on the prediction of falls risks, but more specific approaches based on motion analysis are required to deepen our understanding of particular movement-related falls risks during active walking.

Biomechanical approaches describe a fall as a two-fold scenario: perturbation to dynamic balance and subsequent failure in balance recovery [24–26]. Quantifiability is the strength of biomechanical approaches that can objectively evaluate individuals' falls risks and the effectiveness of intervention strategies (i.e., training, rehabilitation and assistive device) [27]. The current review aims to summarise the latest research findings on fall prevention from biomechanical aspects, but it is important to first highlight the key epidemiologic information about falls in the senior population.

## 2. Fall Prevention for Sustainable Healthcare Systems

Globally, about one in three older adults above 65 years old fall at least once a year and half of them experience multiple falls [28,29]. About 9–20% of the falls lead to serious injuries (e.g., fractures), hospitalisation, visits to the emergency department and death [30–32]. Medical costs due to fall-related injuries are difficult to directly compare between countries, but in a U.S. report, USD 50 billion was the estimated cost in 2018 [33]. For sustainable social security systems in our ageing society, reductions in fall-related injuries are thus urgently demanded because future forecasts predict further acceleration in ageing. In Japan in 2010, the total national revenue was reported to be USD 500 billion, when healthcare costs alone (i.e., medical, nursing care etc.) in the same year exceeded USD 550 billion [2]. This serious financial deficit was compensated for by national bonds to double the budget,

but as a result, Japan has been suffering from the largest amount of national debts in the world (close to USD 9 trillion in 2021) [34]. For such an unsustainable healthcare system to fundamentally reform, it is essential to promote 'healthy life expectancy' with reduced reliance on the conventional 'medical treatment' or 'injury rehabilitation' schemes. In this sense, 'fall prevention' is important for the senior population, and the following section introduces the major direct causes of falls, namely tripping and slipping.

## 3. Biomechanical Factors for Falls Risks

There are a number of factors interacting with each other to increase falls risks. However, it is estimated that 59–78% of falls are due to either tripping or slipping [24,35]. Although a fall can be described as a two-fold episode (i.e., balance perturbation and failure in balance recovery, efforts should first be devoted to the 'prevention of potential balance perturbation'.

### 3.1. Biomechanics of Tripping and Minimum Foot Clearance (MFC)

Tripping, the leading cause of falls, is defined as unexpected foot contact with the walking surface or an object upon that surface, which generates momentum sufficient to destabilise the walker [36]. The two key factors for tripping falls are, therefore, (i) physical foot contact and (ii) excessive momentum. Focusing on these two essential conditions for falls by tripping, minimum foot clearance (MFC) is recognised as the critical gait event (Figure 2) [37].

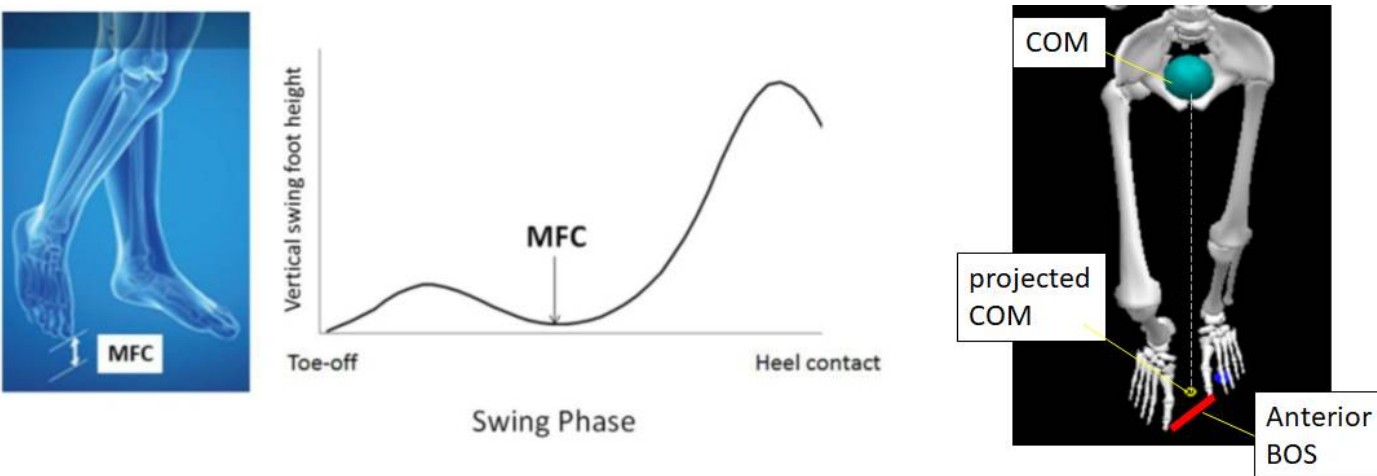

**Figure 2.** (**Left**) Illustration of minimum foot clearance (MFC) and swing foot clearance graph [38]; (**right**) frontal plane illustration at MFC, COM = centre of mass, BOS = base of support, BOS boundary indicating the line between the toes (frontal BOS boundary).

MFC is the mid-swing phase event, where the vertical swing foot displacement reaches the local minimum while travelling near maximum horizontal velocity [39]. The position of both feet is parallel at MFC (Figure 2, right), leaving the small base of support (BOS) vulnerable to forward balance loss and subsequent falls [40,41]. MFC characteristics can, however, be individual-specific, and some older adults do not experience MFC events, possibly due to a slower gait speed and inadequate joint control [42]. The optimisation of MFC control is most important in preventing falls by tripping [37]. While a higher MFC is fundamental, it is also important to achieve constant swing foot control (i.e., intra-individual gait control consistency), indicated by variability measures such as standard deviation (SD) and interquartile range (IQR) [37,43]. As shown in Figure 3 (left), a lower IQR (or SD) indicates the ability for consistent MFC control [44].

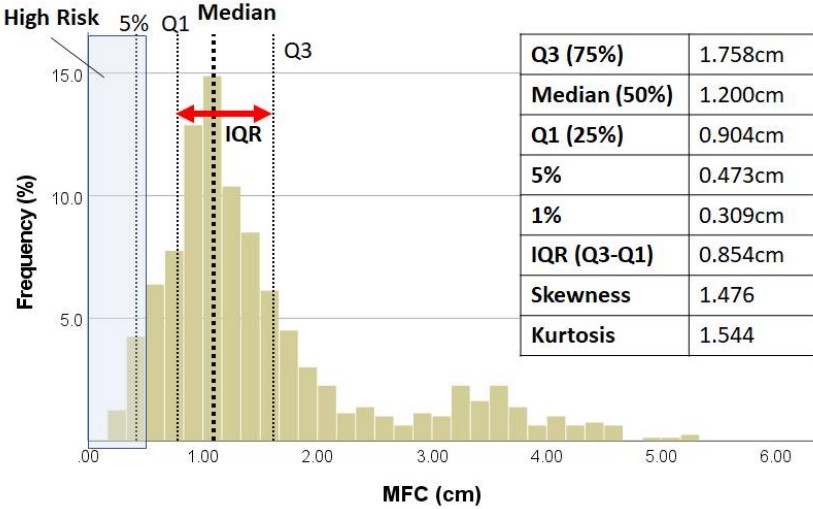
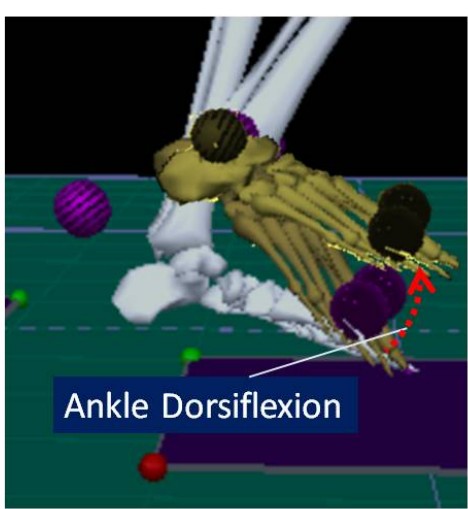

| | |
|---|---|
| Q3 (75%) | 1.758cm |
| Median (50%) | 1.200cm |
| Q1 (25%) | 0.904cm |
| 5% | 0.473cm |
| 1% | 0.309cm |
| IQR (Q3-Q1) | 0.854cm |
| Skewness | 1.476 |
| Kurtosis | 1.544 |

**Figure 3.** (**Left**) MFC histogram description, IQR = interquartile range, Q1 = 25th percentile, Q3 = 75th percentile; (**right**) illustration of ankle dorsiflexion to increase MFC.

MFC can be analysed in detail by plotting the data in the histogram to understand the individual-specific MFC characteristics [37]. As highlighted in Figure 3, the area marked 'High Risk' is of particular importance because the risk of tripping increases at occasional low swing foot clearances. While most previous studies have focused on central tendency and its dispersion, such as mean ± SD or median ± IQR to describe MFC characteristics, for fruitful future directions, lower percentiles (i.e., 0–5%) should be the focus for the most dangerous instances [45].

Despite the importance of a sophisticated analytical approach for MFC data, the fundamental direction is to provide sufficient MFC height, achieved most efficiently by ankle dorsiflexion (Figure 3, right), such that 1° of ankle dorsiflexion elevates the MFC by 0.3cm [46], but ageing is a factor that causes reduced dorsiflexion [47]. Enhanced dorsiflexion at MFC is, therefore, the target for tripping prevention [48,49]

### 3.2. Biomechanics of Slipping and Coefficient of Friction (COF)

Slipping is the second leading cause of falls, but unlike tripping, backward and sideways loss of balance can be consequences, often resulting in the worst type of injury: hip fracture [40]. Downey et al. [50] reported that the one-year mortality of hip fractures is 22% for older individuals.

Biomechanically, slipping can be described as 'horizontal foot velocity during the stance phase' when the foot is, in principle, in contact with the walking surface. After heel contact, the foot stance is not supposed to move anteriorly, but when the foot contact's force outweighs the maximum available friction, the excess anterior force generates acceleration, causing a slip. A slip can be defined by displacement, velocity or acceleration but needs to be distinguished from a micro-slip, up to 3 cm horizontal displacement, following heel contact as part of the regular gait cycle [51].

Friction has the ability to counter-match the shear force, but depending on the interface quality, it has a maximum threshold above which the object (i.e., foot) can start moving. The maximum available friction can be computed as

$$\text{Maximum Friction} = \text{coefficient of friction (COF)} \times \text{Normal Force}$$

where COF indicates 'slipperiness of the interface' and Normal Force is the GRF component perpendicular to the walking surface, therefore, the vertical GRF component in level walking. As long as the maximum available friction is higher than the horizontal foot contact force, a slip can be prevented, but in level walking, the alternative condition is 'COF is higher than the certain threshold, known as the required coefficient of friction

(RCOF)' [52,53]. The RCOF is determined by walking patterns, more specifically, the ratio of GRF between horizontal and vertical components.

$$RCOF = \left| \frac{GRFhor}{GRFver} \right|$$

Throughout the stance phase of the gait cycle, GRF characteristics continuously change, but the initial peak RCOF is the marker for the risk of slipping widely varying between individuals and walking patterns. Immediately following heel contact, the RCOF values are often 'noisy' (Figure 4), usually disregarded from the slip-risk assessment. This noisy part of RCOF probably reflects a micro-slip, a small movement of the heel (i.e., up to 3 cm) immediately following foot contact [51]. In short, a slip-resistant interface (i.e., shoe-outsole and floor treatment) is required against the peak RCOF after heel contact (heel slip risk in Figure 4).

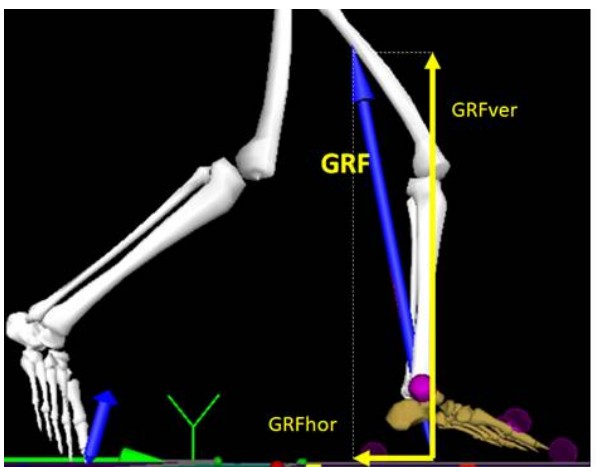 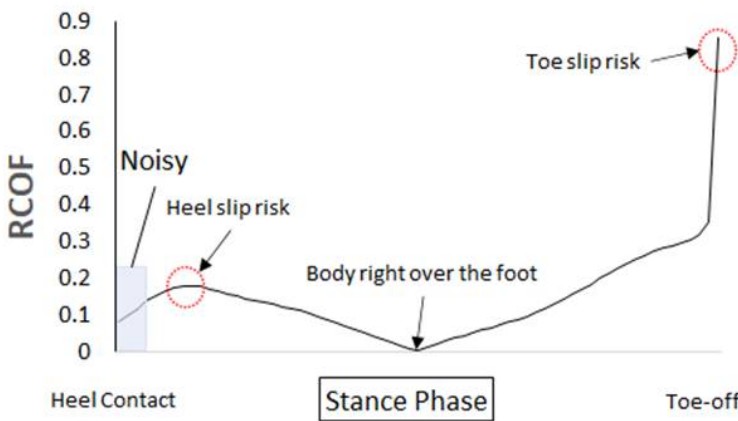

**Figure 4.** (**Left**) Characteristics of ground reaction forces (GRF) in the sagittal plane, with GRFver/hor indicating vertical/horizontal components, respectively; (**right**) required coefficient of friction (RCOF) characteristics throughout the stance phase.

## 4. Biomechanics of Dynamic Balance

Even after tripping or slipping, falls can still be prevented as long as balance is restored. 'Balance' has been commonly used in many everyday contexts, but it was not until recently that human balance was properly quantified based on biomechanical concepts. In the simplest definition, balance is secured when COM is within BOS in the transverse plane [26]. This fundamental definition holds true as long as the body remains static, ensuring absolute safety. This concept is, however, not practical in predicting falls risks. First, BOS in bipedal human locomotion is defined as the foot contact area, plus the area between the two feet during the double support phase [54]. Following this balance definition, human gait can be viewed as a continuum of 'balance loss and balance recovery' by linking the single- and double-limb supporting phases alternately [55]. It is, however, obvious that balance loss during the single-support phase (when COM is outside BOS) is not necessarily related to falls, but it can be rather functional for the purpose of walking (i.e., forward progression). Hazardous balance loss needs to be clearly distinguished from functional requirements. This issue can be overcome by considering virtual (projected) BOS that has been introduced by Patla et al. [26], in that both foot positions in the transverse plane define the area regardless of whether the foot is in contact with the walking surface (Figure 5). A balance assessment can be relative to this virtual BOS to characterise a hazardous balance loss that leads to falls. Second, in dynamic situations such as human walking, velocity needs to be considered rather than looking only at positional relationships between COM and BOS. Even in the same positional relationships, COM velocity can be a large determinant for balance conditions in dynamic situations.

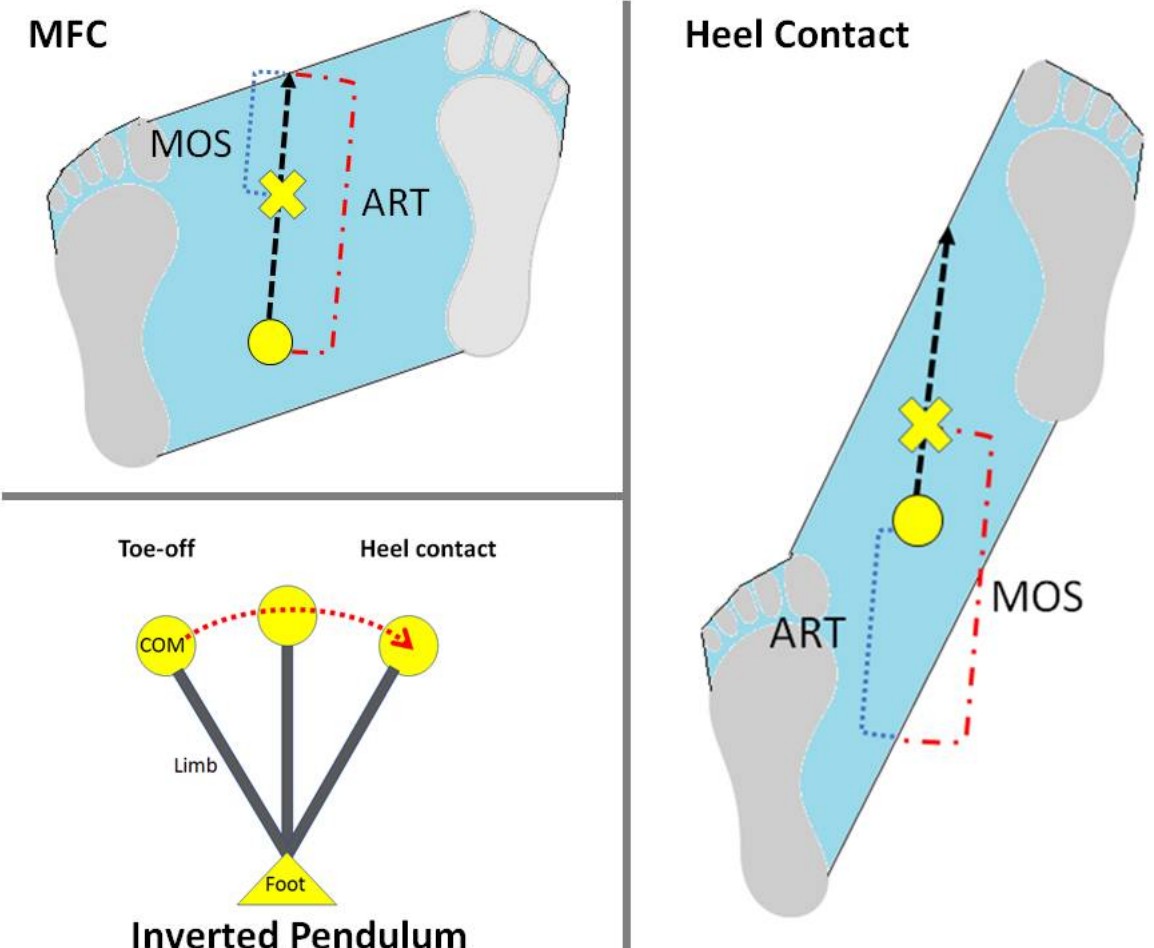

**Figure 5.** (**Top left**) Dynamic balance characteristics at MFC—tripping-related COM indicated by a circle, XCOM indicated by X, (**bottom left**) inverted pendulum model during single support; (**right**) dynamic balance characteristics at heel contact—slipping related; MOS = margin of stability COM indicated by a circle, XCOM indicated by X. ART = available response time, COM = centre of mass. XCOM = COM + COMvelocity + $\sqrt{g/l}$, where g = gravitational acceleration and l = limb length; MOS = BOS—XCOM as illustrated above.

### 4.1. Margin of Stability (MOS)

A spatial description of dynamic balance can be provided by the margin of stability (MOS), as illustrated in Figure 5. The fundamental concept of MOS is the distance between the extrapolated centre of mass (XCOM) and the BOS boundary relevant to the particular balance loss type.

$$MOS = BOS - XCOM$$

In the calculation of dynamic balance, XCOM is the projected COM position based on current velocity factors, therefore, differentiating 'moving' COM from the static condition. Falls direction can be determined by which BOS boundary COM is likely to cross. Tripping and slipping are the primary causes of forward and backward balance loss, respectively [40]. Although the MOS concept may appear to ignore vertical COM information, the calculation of XCOM requires the limb length, accounting for the vertical kinematic factor for dynamic balance. As in Figure 5 (left bottom), inverted pendulum gait mechanics [55] account for the arch-like COM motion path.

$$XCOM = COM + \frac{COMvel}{\sqrt{\frac{g}{l}}}$$

Originally, the most traditional concept for dynamic balance was viewed in the transverse plane (Figure 5) [55] with resultant anterior–posterior and medio–lateral components together. However, MOS studies do not focus on actual (resultant) COM movement in the transverse plane due to attempts to conduct sagittal and/or frontal plane analyses [56–60]. Viewing MOS in the sagittal or frontal plane, however, masks the actual COM movement direction and has absolutely no advantages over a transverse plane analysis. Even when there is sufficient anterior MOS, for example, balance can still be destabilised in the medio–lateral directions and vice versa. MOS should thus be considered in the transverse plane (Figure 5) and utilising the actual COM direction but not in the sagittal or frontal plane.

*4.2. Available Response Time (ART)*

In addition to the spatial definition of dynamic balance, available response time (ART) is used to describe the temporal balance. The concept for computing the estimated time of COM to cross the BOS boundary is simple.

$$\mathrm{ART} = \frac{(\mathrm{BOS} - \mathrm{COM})}{\mathrm{COMvel}}$$

Again, an important consideration is to identify the relevant BOS boundary and balance loss directions [41,61].

## 5. Fall Prevention Strategies

Gait biomechanics attributes age-associated falls risks to changes in gait patterns, most fundamentally recognised as a slower walking speed, a shorter step length, a larger step width and prolonged double support time [38]. Ageing also negatively alters various neuromotor systems, such as a slower reaction and the loss of fine-movement control [62,63], all of which are combined together to increase the likelihood of falls. For older adults, Yardley et al. [64] reported that fall prevention strategies should be easy to implement and affordable, immediately affecting and supporting active walking; otherwise, they would not be followed for a long time even if they were effective. This section introduces fall prevention strategies that may cover all of these aspects: footwear, exercise and technology-based interventions.

*5.1. Footwear Intervention*

Footwear is an essential consideration for fall prevention, with various potential effects on reducing the risk of tripping and slipping. In short, tripping prevention can focus on increasing swing foot clearance (i.e., MFC) by modifying the footwear geometry, while shoe-outsole modifications can provide sufficient friction to minimise slipping. Menant and colleagues [65,66] investigated the effects of shoes on gait biomechanics and associated falls risks. Their research outcomes were largely in agreement with previous reports about 'safe footwear'.

A shoe insole is a direct interface for the foot and influences gait control mechanics. Nagano and Begg [49] introduced ISEAL technology, which supports ankle dorsiflexion and eversion, especially in the heel. As in Figure 6 (top left), the specific heel geometry combined with texture installation succeeded in increasing MFC and guiding the foot's centre of pressure (COP) to stabilise balance [38]. Maki et al. [67] devised a shoe insole called SoleSensor©, installing tubes peripherally to trigger cutaneous receptors when the foot COP travels toward the BOS boundary and to promote afferent feedback for faster reaction speed to avoid balance loss. Another type of shoe insole consideration is the incorporation of additional cushioning effects to reduce plantar foot pressure. Custom moulding is also common for shoe insoles, often utilising 3D foot scanning systems to maximise the contact area between the foot and the shoe insole. Such foot-pressure-redistribution techniques may not directly reduce falls risks but can protect the foot from deformity and pain, possibly helping to sustain dynamic balance. The custom moulding of shoe insoles, however, may not always solve foot problems but, rather, could exacerbate the condition. If a foot

deformity exists, merely scanning the foot results in moulding the shape of the damaged foot and makes it rather difficult to overcome these problems.

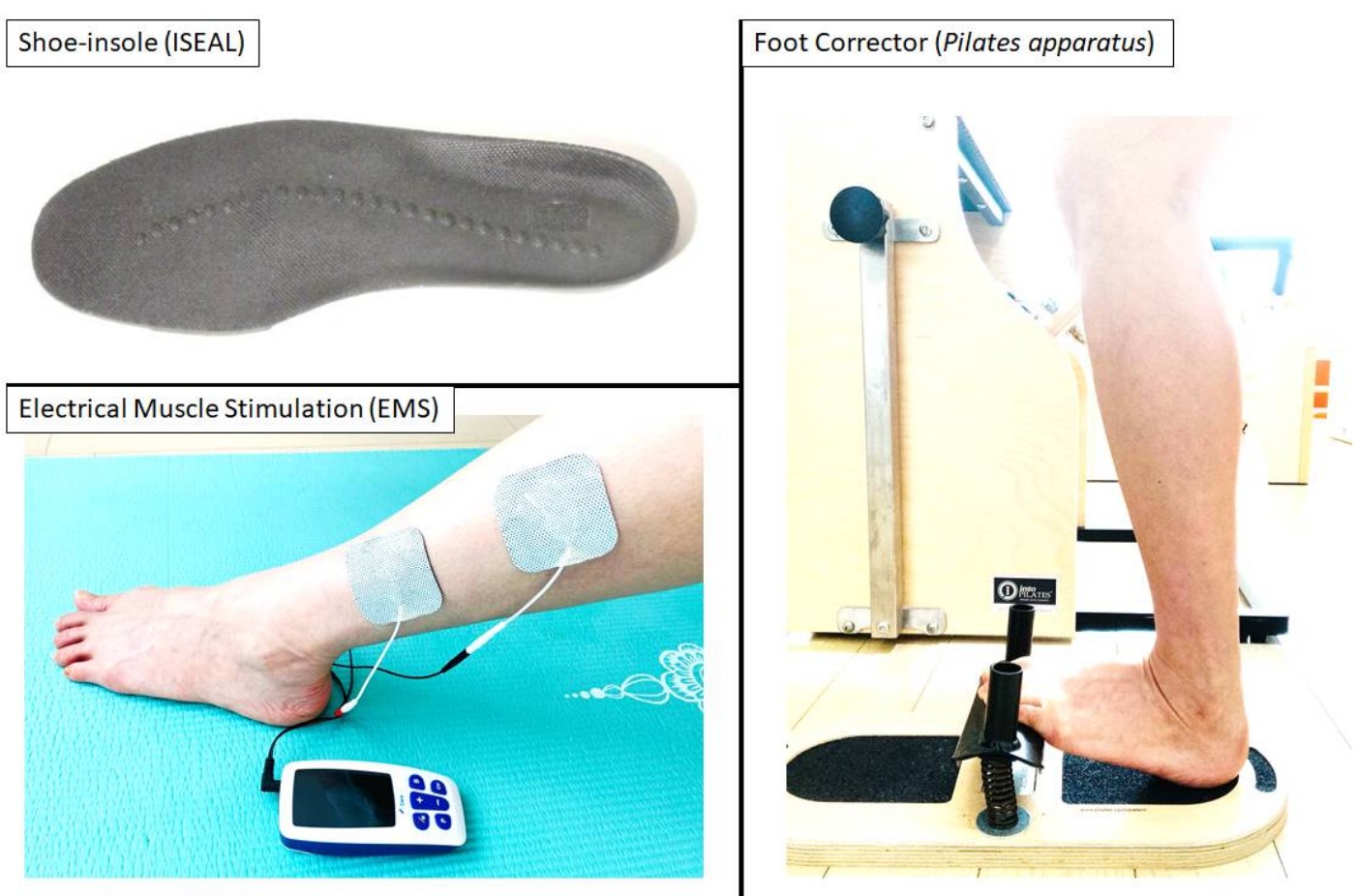

**Figure 6.** (**Top left**) Texture installation on the insole to guide centre of pressure movement; (**bottom left**) application of electrical muscle stimulation on dorsiflexors; (**right**) foot corrector for dorsiflexors' training.

Contrary to a shoe-insole intervention to reduce the risk of tripping, slip prevention can be summarised as 'COF greater than RCOF', but attempts to reduce RCOF may not be practical. A lower RCOF can be achieved with smaller horizontal and greater vertical GRF components. Such walking patterns can be characterised by 'stomping' but cannot be recommended as everyday gait adaptations due to potential damages to lower limb joints. Shoe modification for slip prevention is, for this reason, almost limited to the outsole structure by providing sufficient friction.

For slip prevention, the horizontal foot contact force must be smaller than the friction between the two interfaces: the walking surface and shoe outsole. The indicator of 'slipperiness' and coefficient of friction (COF) is determined mostly by materials and surface conditions (i.e., contamination), but there is no single best outsole material for reducing the risk of slipping. Polyurethane (PU) is, for example, slip-resistant, particularly against a contaminated surface compared to other materials, but effects can wear off relatively in a short period of time [68,69]. Thermal plastic rubber (TPR) is generally not slip-resistant; however, TPR can be useful on an icy surface. On a dry surface, natural rubber is the optimal material, but it is questionable whether we need to consider falls from slipping when the surface is not slippery. Softer outsole materials generally provide higher COFs but can wear off and lose their slip resistance quicker. Considering the lifespan of footwear, extremely soft outsoles may not be cost-effective. Outsole treads can be another consideration, but further research should be undertaken to effectively reduce the risk of slipping [70].

Footwear intervention potentially has various advantages, but there are a few cautions to consider because many types of commercially available footwear could possibly cause adverse effects on foot health and safe walking. For example, shoe insoles with elevated heels are commonly available on the market but can cause foot pressure to concentrate on the metatarsal regions and could eventually deform the foot [38]. As with high heels, an elevated heel is not optimal for everyday walking [65]. Second, asymmetrical footwear is difficult to justify without appropriate fitting and expert advice. The fundamental concept of footwear intervention should aim for a symmetrical gait style, but some prefabricated commercial footwear products, including shoe insoles and shoe outsoles, have incorporated asymmetrical features, which could potentially cause various lower-limb joint problems with long-term use.

### 5.2. Exercise Intervention

Exercise intervention has been reported to be effective at enhancing seniors' health and at reducing falls risks [71]. It is, however, important to focus on motivational issues that are related to individuals' health statuses. Without sufficient physical, psychological and social health, older adults are unlikely to engage in exercise continuously [72,73]. For this reason, factors for encouraging voluntary participation must be considered, such as ease of engagement, some sort of immediate effects and cost affordability [60].

Exercise prescription generally follows the same principle as that of the younger populations. Muscle development is, for example, based on the 'overload principle' and 'supercompensation' [74]. Differences, however, exist in various physiological conditions such as sarcopenia, bone density, maximal aerobic capacity and pathological conditions (e.g., cardiovascular dysfunctions, hypertension, diabetes, etc.) [75,76]. It is essential to account for age-associated health declines to prevent adverse results of exercise for older adults.

As repeatedly emphasised, dorsiflexion is the key ankle motion that provides sufficient swing foot clearance at MFC and prevents tripping [46]. In addition, dorsiflexion supports 'defined' heel contact at the beginning of a stance phase, which is effective at reducing foot contact impact [77]. Tibialis anterior is the primary muscle activating ankle dorsiflexion, and exercise intervention should, therefore, focus on dorsiflexor muscle groups. Figure 6 (right) introduces the use of an apparatus called 'foot corrector' that can effectively train both the dorsiflexors and the plantarflexors.

Another important element of exercise intervention is so-called 'core training' to stabilise bodily COM by toning the muscles around the pelvis–trunk region [78,79]. In this concept, the strengthening of trunk stabilisers can help COM to be steadily positioned within BOS.

### 5.3. Technology-Based Intervention

A number of technology-based interventions are available, but one promising method is electrical stimulation, which can enhance the necessary muscles through improved neuromotor control [80,81]. Electrical stimulation is commercially available as electrical muscle stimulation (EMS) or transcutaneous electrical nervous stimulation (TENS). These portable devices trigger motor neuron excitation and their associated muscle contractions via electrical signals. Depending on the system, stimulation patterns, including intensity, frequency and rhythm, can be controlled to provide the optimal effects [82]. There are many advantages to electrical stimulation, including (i) cost-effectiveness, (ii) passive treatment option—not requiring physical burdens and being suitable for frail populations, and (iii) few known adverse side effects (e.g., excessive use not being recommended and allergic reactions to electrodes' gel in rare cases). There are unlimited potentials in this application, and much is still to be uncovered, but a straightforward application is attachment to the tibialis anterior (i.e., dorsiflexors), as in Figure 6 (bottom left). As explained above, strengthening the dorsiflexors could increase the swing foot clearance and reduce the risk of tripping [46].

This mechanism has been incorporated into some active orthoses utilising a footswitch to activate the electrical stimulation of dorsiflexors. In a previous report [83], a footswitch was installed at the heel to detect heel contact and heel off, providing electrical stimulus to the tibialis anterior from heel off until heel contact. In a healthy gait, however, the dorsiflexors should be active primarily during the swing phase, from toe off to heel contact [84]. Multiple footswitches should be, therefore, installed on both the heel and toe to detect heel contact and toe off. It is also important to note that dorsiflexors have two peaks: slightly before mid-swing and immediately after heel contact [85]. The intensity of electrical stimulation on the tibialis anterior should therefore be controlled to assist with achieving optimal ankle kinetic requirements [86,87].

Another type of active system is the wearable cyborg (active exoskeleton) such as Hybrid Assistive Limb™ (HAL™, Prof. Sankai University of Tsukuba/CYBERDYNE Inc.). It is a motor-driven exoskeleton system to produce the wearer's intended motions through the actuation of HAL's motors by converting efferent neuro-signals, whether erratic or faint, into tangible movements that meet the intention of its wearer through processing. (Figure 7) [88]. While this type of wearable cyborg can assist people while it's being worn, it can re-establish the lost connection between the efferent motor signals and the afferent sensory signals of the body's ori ginal biofeedback loop of patients with severe movement dysfunctions (e.g., spinal cord injury, post-stroke syndromes), by acting as a non-invasive external medium through coupling with the 'intention movement' of the wearer (i.e., interactive biofeedback theory). While this theory was proven through several clinical trials, Further research is required to test whether the use of wearable cyborgs provides learning effects of lower limb control and reduces falling risks.

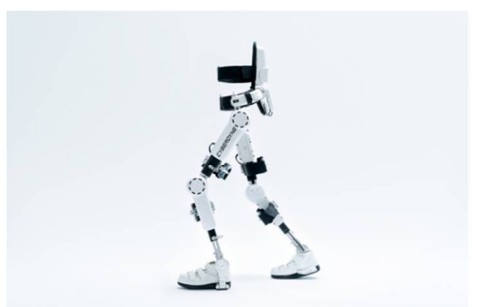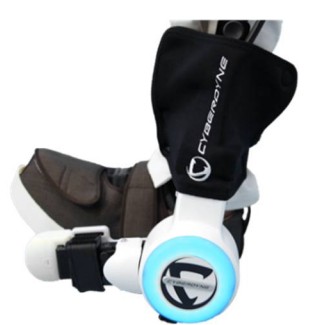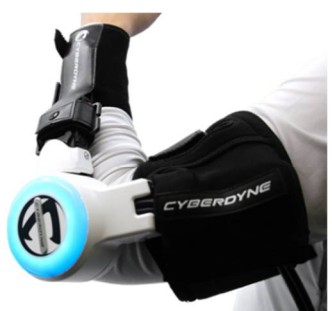

**Figure 7.** Hybrid Assistive Limb (HAL™, Prof. Sankai, CYBERDYNE Inc.), (**left**) Lower Limb Type; (**middle**) Single Joint Type- ankle joint attachment; (**right**) Single Joint Type- elbow joint.

## 6. Conclusions

Fall prevention is urgently required for a sustainable ageing society, and biomechanical approaches identified tripping and slipping as the two primary causes of hazardous balance loss. Minimum foot clearance (MFC) and the required coefficient of friction (RCOF) are key parameters used to predict the risk of tripping and slipping, respectively. Controlling the centre of mass and establishing an effective base of support can be useful for maintaining dynamic balance, leading to fall prevention. Dynamic balance can be quantified by a margin of stability (MOS) and available response time (ART) to assess individuals' capacity for balance recovery. Various intervention strategies for fall prevention include optimum footwear, adequate exercise and technology-based approaches. A biomechanical understanding of falls among older adults can be applied to practical solutions to reduce falls risks. While a biomechanical understanding of each element is important, it is first necessary to picture the overall framework of gait analysis for fall prevention among older adults.

**Funding:** This research received no external funding.

**Conflicts of Interest:** The author declares no conflict of interest.

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
