# Peer review of "Gait Biomechanics for Fall Prevention among Older Adults"

_applsci, doi:10.3390/app12136660_

Round 1

Reviewer 1 Report

I think this paper and its contents are interesting.

Author Response

We appreciate the reviewer's positive comment.

Reviewer 2 Report

The authors reviewed gait biomechanics of falls among older adults. Biomechanical factors for falls risks (tripping and slipping) and biomechanics of dynamic balance as well as strategies to prevent falls were discussed. The content of this review is valuable and attractive. The manuscript was well written, so I suggest this work be published in Applied Sciences in its current manner.

Author Response

Thank you for taking your time to review the manuscript.

Reviewer 3 Report

The review article entitled “Gait Biomechanics for Falls Prevention among Older Adults” focus a very general topic that, although potentially of relevance, does not seem in line with the scope of this Special Issue "Joint Kinematics Analysis and Injuries Recovery". Specifically, the manuscript is focused on fall prevention, and not injury recovery, and does not present a comprehensive review of joint kinematics related to falling.

Further, the review presents several methodological flaws:

1.       The aim is too general (“The current review aims to summarise latest research findings about falls prevention from biomechanical aspects, but it is first important to highlight the key epidemiologic information about falls in the senior population”) with no specific questions to address.

2.       There is no information about the methodology followed to perform this review.

3.       The subtopics are not comprehensive, excluding several important references, specially regarding gait changes that are associated to falling and exercise interventions recommended to prevent falls.

4.       The conclusion is also very general and does not add a valuable contribution for the topic.

Author Response

There seemed to be some mistakes in administration because the current manuscript has been submitted to the Special Issue, “Falls: Risk, Prevention and Rehabilitation” but not “Joint Kinematics Analysis and Injuries Recovery”.

  1. The aim is too general (“The current review aims to summarise latest research findings about falls prevention from biomechanical aspects, but it is first important to highlight the key epidemiologic information about falls in the senior population”) with no specific questions to address.

Reply: Section 2 summarises the epidemiologic information including frequency, injury rate and medical costs of falls-related injuries. In addition, the section talks about direct causes of falls etc. The primary purpose of the current review is to provide the biomechanical framework for falls prevention among older adults but not to answer specific research questions. This point has been incorporated into the revised manuscript as follows. (Line 379-380)

“While biomechanical understanding of each element is important, it is first necessary to picture the overall framework of gait analysis for falls prevention among older adults.”

  1. There is no information about the methodology followed to perform this review.

Reply: The current manuscript is not a systematic review because it does not focus on a few controversial issues that need clarification based on previous research studies. Instead, the manuscript is meant to be positioned as the reference point for the overall conceptual framework of gait biomechanics for falls prevention among older adults. Please see the comment above relating to the conclusion (Line 379-380).

  1. The subtopics are not comprehensive, excluding several important references, specially regarding gait changes that are associated to falling and exercise interventions recommended to prevent falls.

Reply: Further information about age-specific adaptations to spatio-temporal gait parameters have been included in the revised manuscript. (Line 235-239)

“Gait biomechanics attributes age-associated falls risks to changes in gait patterns, most fundamentally recognised as slower walking speed, shorter step length, larger step width and prolonged double support time [38]. Ageing also negatively alters various neuromotor systems such as slower reaction and loss of fine-movement control [62, 63], all of which are combined together to increase the likelihood of falls.”

Further information about exercise intervention have been incorporated into the revised manuscript as follows. (Line 309-320)

“Exercise intervention has been reported to be effective for enhancing the seniors’ health and reducing falls risks [71]. It is, however, important to focus on motivation is-sues that are related to individuals’ health statuses. Without the sufficient physical, psychological and social health, older adults are unlikely to engage into exercise con-tinuously [72, 73]. For this reason, factors to encourage voluntary participation must be considered such as ease of engagement, some sorts of immediate effects and cost af-fordability [60].

Exercise prescription generally follows the principle same as that for younger populations. Muscle developments are, for example, based on ‘overload principle and supercompensation [74]. Differences, however, exist in various physiological conditions such as sarcopenia, bone density, maximal aerobic capacity and pathological conditions (e.g. cardiovascular dysfunctions, hypertension, diabetes, etc) [75, 76]. It is essential to account for age-associated health declines to prevent adverse results of exercise for older adults.”

  1. The conclusion is also very general and does not add a valuable contribution for the topic.

Reply: Conclusion section has been revised as follows. (Line 369-380)

“Falls prevention is urgently required for sustainable ageing society and biomechanical approaches identified tripping and slipping as the two primary causes of hazardous balance loss. Investigation of minimum foot clearance (MFC) and required coefficient of friction (RCOF) are the key parameters to predict the risk of tripping and slipping, respectively. Centre of mass control and effective base of support establishment can be useful for maintaining dynamic balance, leading to falls prevention. Dynamic balance can be quantified by margin of stability (MOS) and available response time (ART) to assess individuals’ capacity for balance recovery. Various intervention strategies for falls prevention include footwear, exercise and technology-based approaches. Biomechanical understanding of falls among older adults can be applied into practical solution to reduce falls risks. While biomechanical understanding of each element is important, it is first necessary to picture the overall framework of gait analysis for falls prevention among older adults. ”

Reviewer 4 Report

In this paper, the author discusses about gait biomechanics of elderly people, presenting several techniques to minimize the risk of falling or slipping.

Although English level is generally good enough to easily understand the article, some parts of the text use quite unnatural phrase constructions, and some others lack punctuation marks (i.e. L28 to L31 without a single comma). In my humble opinion, the text may benefit from a review by a native English speaker.

The article is well structured, and it may appeal any reader interested in the topic. However, it does not add any new information to the research field, and it limits to highlight just some of the findings of other researchers. In my opinion, the article would boost its interest if it was oriented as “systematic research”.

I have some comments that the author may want to consider to improve the article:

- L70-72 - If the data refers to worldwide population, it should be highlighted, otherwise reader may think that it refers to Japanese population (as stated before in Introduction)

- Figure 2 - Low image resolution make some parts of the figure blurry, this should be revised and improved. Also, floating "spheres" size make difficult to appreciate MFC, BOS and COM definition. I would advise to reduce them, or to use an alternative figure with clearer reference planes to define them.

Also, standing foot (left) appear with a weird angle, like if the walker was wearing heels, and no contact can be appreciated with the reference floor plane. This can be confusing and misleading.

- L103-106 gait pattern is not universal and is quite age-dependent, as it is well known it changes with gender and age. Young people's feet land over the heel, while elderly people land with the sole, using shorter steps, something that also has an influence. This should be at least mentioned in main text, and reference to this should be included, in my opinion.

- First reference to MOS and XCOM (L182) appear in Figure 5, before they are defined within text (L196-198). I would recommend adding a brief definition of them in Figure 5 caption, along with the rest of abbreviations, to ease reading.

- Conclusions section is very short, I would advise to increase its length and highlighting possible future development lines in this field.

Author Response

While the reviewer may not think there are much new information in the current review manuscript, I would like to clarify that it was meant to be positioned as the conceptual framework for falls prevention based on gait biomechanics. In addition, review papers generally aim to summarise the previous studies rather than delivering new information based on experimental studies. There have been various previous research papers detailing specific topics. For example, there are so much details in biomechanics of tripping, slipping and balance. However, I have barely read papers that provide a conceptual framework to include all these specific details under the same umbrella of falls prevention among the older population. The manuscript can, therefore, be the reference point to understand the overall picture of biomechanical approaches for falls prevention among the older population.

- L70-72 - If the data refers to worldwide population, it should be highlighted, otherwise reader may think that it refers to Japanese population (as stated before in Introduction)

Reply: Japan was mentioned as the example of the most aged country. As per the reviewer’s comment, information relating to the global population has been highlighted as below. (Line 70-71)

“Globally, about one in three older adults above 65 years old fall at least once a year and half of them experience multiple falls [28, 29].”

- Figure 2 - Low image resolution make some parts of the figure blurry, this should be revised and improved. Also, floating "spheres" size make difficult to appreciate MFC, BOS and COM definition. I would advise to reduce them, or to use an alternative figure with clearer reference planes to define them.

Also, standing foot (left) appear with a weird angle, like if the walker was wearing heels, and no contact can be appreciated with the reference floor plane. This can be confusing and misleading.

Reply: Figure 2 has been re-developed.

- L103-106 gait pattern is not universal and is quite age-dependent, as it is well known it changes with gender and age. Young people's feet land over the heel, while elderly people land with the sole, using shorter steps, something that also has an influence. This should be at least mentioned in main text, and reference to this should be included, in my opinion.

Reply: This point in relation to MFC has been incorporated in the revised manuscript as follows. (Line 108-110)

‘MFC characteristics can be, however, individual specific and some older adults did not have MFC events possibly due to slower gait speed and inadequate joint control [42].’

- First reference to MOS and XCOM (L182) appear in Figure 5, before they are defined within text (L196-198). I would recommend adding a brief definition of them in Figure 5 caption, along with the rest of abbreviations, to ease reading.

Reply: Further explanation has been added to Figure 5 caption as follows.

“XCOM = COM + COMvelocity + √g/l, where g = gravitational acceleration and l = limb length; MOS = BOS – XCOM as illusatrated above.”

- Conclusions section is very short, I would advise to increase its length and highlighting possible future development lines in this field.

Reply: Conclusion has been revised as follows. (Line 369-380)

“Falls prevention is urgently required for sustainable ageing society and biomechanical approaches identified tripping and slipping as the two primary causes of hazardous balance loss. Investigation of minimum foot clearance (MFC) and required coefficient of friction (RCOF) are the key parameters to predict the risk of tripping and slipping, respectively. Centre of mass control and effective base of support establishment can be useful for maintaining dynamic balance, leading to falls prevention. Dynamic balance can be quantified by margin of stability (MOS) and available response time (ART) to assess individuals’ capacity for balance recovery. Various intervention strategies for falls prevention include footwear, exercise and technology-based approaches. Biomechanical understanding of falls among older adults can be applied into practical solution to reduce falls risks. While biomechanical understanding of each element is important, it is first necessary to picture the overall framework of gait analysis for falls prevention among older adults. ”

Round 2

Reviewer 4 Report

All my previous comments have been considered.